# Investigating Novel IspE Inhibitors of the MEP Pathway in Mycobacterium

**DOI:** 10.3390/microorganisms12010018

**Published:** 2023-12-21

**Authors:** Seoung-Ryoung Choi, Prabagaran Narayanasamy

**Affiliations:** Department of Pathology and Microbiology, College of Medicine, University of Nebraska Medical Center, Omaha, NE 68198, USA

**Keywords:** pathogens, IspE inhibitor, carboline, synthesis, minimum inhibitory concentration

## Abstract

In a recent effort to mitigate harm from human pathogens, many biosynthetic pathways have been extensively evaluated for their ability to inhibit pathogen growth and to determine drug targets. One of the important products/targets of such pathways is isopentenyl diphosphate. Isopentenyl diphosphate is the universal precursor of isoprenoids, which are essential for the normal functioning of microorganisms. In general, two biosynthetic pathways lead to the formation of isopentenyl diphosphate: (1) the mevalonate pathway in animals; and (2) the non-mevalonate or methylerythritol phosphate (MEP) in many bacteria, and some protozoa and plants. Because the MEP pathway is not found in mammalian cells, it is considered an attractive target for the development of antimicrobials against a variety of human pathogens, including *Mycobacterium tuberculosis* (*M.tb*). In the MEP pathway, 4-diphosphocytidyl-2-c-methyl-d-erythritol kinase (IspE) phosphorylates 4-diphosphocytidyl-2-C-methyl-D-erythritol (CDPME) to form 4-diphosphocytidyl-2-C-methyl-D-erythritol 2-phosphate (CDPME2P). A virtual high-throughput screening against 15 million compounds was carried out by docking IspE protein. We identified an active heterotricyclic compound which showed enzymatic activity; namely, IC_50_ of 6 µg/mL against *M.tb* IspE and a MIC of 12 µg/mL against *M.tb* (H37Rv). Hence, we designed and synthesized similar new heterotricyclic compounds and tested them against mycobacterium, observing a MIC of 5 µg/mL against *M. avium*. This study will provide the critical insight necessary for developing novel antimicrobials that target the MEP pathways in pathogens.

## 1. Introduction

Pulmonary tuberculosis (TB), caused by *Mycobacterium tuberculosis* (*M.tb*), is a worldwide health problem, with over 2 billion people infected with latent TB globally. About 450,000 people a year are infected with multi-drug-resistant *M.tb* (MDR-TB), which is resistant to the main first-line drugs, isoniazid and rifampicin. Additionally, extensively drug-resistant *M.tb* (XDR-TB)—which is resistant to isoniazid, rifampin, all fluoroquinolones, and at least one of three injectable second-line drugs (i.e., amikacin, kanamycin, or capreomycin)—has been reported in 37 countries in all regions of the world since 2006. Human immunodeficiency virus–tuberculosis (HIV-TB) co-infection has contributed in part to the development of this growing pattern of drug resistance. Thus, designing and developing new anti-TB drugs is critical [1].

In contemporary drug discovery research, the quest for effective strategies to combat the deleterious impact of human pathogens remains an area of paramount importance. This pursuit involves a comprehensive exploration of various biosynthetic pathways, evaluating their potential to not only inhibit pathogen growth but also to serve as viable targets for drug development. At the heart of these investigations lies a critical focus on isopentenyl diphosphate, a key precursor indispensable for the synthesis of isoprenoids—a class of compounds which play a pivotal role in the normal functioning of microorganisms.

Biosynthetically, two principal pathways contribute to the production of isopentenyl diphosphate: the well-established mevalonate pathway and the non-mevalonate, or methylerythritol phosphate (MEP) pathway (Figure 1). The emphasis on isopentenyl diphosphate stems from its central role as a universal building block for isoprenoids, which are essential for processes such as cell membrane structure, electron transport, and the biosynthesis of various biomolecules.

A focal point of intrigue within these investigations is the MEP pathway, notable for its absence in mammalian cells. This unique feature positions the MEP pathway as an attractive and, importantly, a differentially exploitable target for developing antimicrobials. The absence of the MEP pathway in humans opens a strategic avenue for designing selective inhibitors, presenting a promising prospect for therapeutic intervention against a broad spectrum of pathogens.

The development of inhibitors targeting the MEP pathway is a dynamic and evolving area of research. Researchers are delving into the intricacies of this pathway, focusing on enzymes that play pivotal roles in isopentenyl diphosphate production. By scrutinizing these key enzymes, novel compounds are being systematically explored with the goal of disrupting the MEP pathway’s functionality. The rationale behind this approach is to impede the synthesis of essential isoprenoids, thereby compromising the normal functioning of microbial organisms.

The MEP pathway has historically been believed to be the pathway whereby microorganisms, including *M.tb*, *M. avium*, and *M. abscessus*, synthesize isoprenoids, which are essential for the growth of these and many other prokaryotes. The MEP pathway’s absence in humans, coupled with its presence in various human pathogens such as *Mycobacterium*, *Pseudomonas*, *Klebsiella*, *Toxoplasma* and *Plasmodium* species, renders its enzymes attractive targets for the design and development of novel antimicrobials. Hence, intensive research has been spurred by this MEP pathway on a worldwide scale. Thus, inhibitors of the MEP pathway are being explored as potential anti-infective agents. Most importantly, IspE has been identified as a better target for determining a critical anti-TB drug. However, recent work [2,3,4,5] has identified that parallel and overlapping steps may be present in the MEP pathway utilized by *M.tb*. We have also shown that IspC [6], IspD [7,8], IspE [9,10], IspF [11] are able to synthesize methyl erythritol cyclo-diphosphate (MECPP) [12] (Figure 1). Recently, our team conducted a detailed study on the MEP pathway, and published purification and kinetic studies of IspC [13], IspD [9], IspE [10] and IspF [11]. The second enzyme in the MEP pathway, IspC, catalyzes a two-step reaction consisting of the Mg^2+^-triggered rearrangement of DXP into a non-isolable aldehyde and the NADPH-dependent reduction of the aldehyde to MEP (Figure 1). Literature on numerous IspC protein crystal structures from various organisms, including important pathogens such as *E. coli*, *M. tuberculosis* and *Y. pestis*, is available. Moreover, the active site of IspC has been identified as a particularly druggable pocket, identifying IspC as a potential target for drug development. For example, fosmidomycin was found to exhibit antibacterial activity against a broad-spectrum Gram-negative bacteria, and antimalarial activity against *Plasmodium* species by targeting IspC. However, the enzyme’s significant conformational changes upon ligand binding challenge the development of novel IspC inhibitors [9,11,12,14].

The third enzyme in the MEP pathway, IspD, serves as a cytidyl transferase, catalyzing the transfer of the cytidyl phosphate group from cytidine triphosphate (CTP) to the phosphate of MEP (Figure 1). This process results in the formation of 4-diphosphocytidyl-2-C-methyl-d-erythritol (CDP-ME) and inorganic diphosphate. Developing inhibitors that are substrate-competitive for IspD presents a challenge due to its active site being predominantly solvent-exposed and the least lipophilic among all the enzymes within the MEP pathway. Additionally, the existence of a homologous cytidyl transferase in human cells raises concerns about the potential application of discovered inhibitors in therapy. Existing literature on IspD inhibition primarily focuses on herbicides and antimalarials [9,11,12,14].

The IspD product, CDPME, undergoes ATP-dependent phosphorylation, resulting in the formation of 4-diphosphocytidyl-MEP (CDP-ME2P). This phosphorylation reaction is catalyzed by a cytoplasmic magnesium ion-dependent kinase IspE (Figure 1). It has been proposed that distinctive features of the IspE active site could facilitate selective targeting. The high conservation of amino acids in the active site provides an opportunity for the design of broad-spectrum antibacterials [9,11,12,14].

IspF, the fifth enzyme in the MEP pathway, catalyzes the formation of 2-C-methyl-d-erythritol-2,4 cyclodiphosphate (MEcPP) (Figure 1). The catalytic mechanism of IspF involves the coordination of two metal cations (Zn^2+^ and Mg^2+^ or Mn^2+^), presenting good opportunities for targeted drug design. The Zn^2+^ ion plays a crucial role by coordinating the phosphate group, thereby enhancing its electrophilicity and facilitating a nucleophilic attack. The intermediate undergoes cyclization, resulting in the formation of MEcPP and cytidine monophosphate (CMP). There are two potential pockets where inhibitors can bind. The first is the active site of IspF, characterized by an unusually high proportion of apolar amino acids. The second is allosteric sites, which open up another dimension for drug development. The allosteric sites, where larger substrate analog inhibitors can bind, offer an alternative strategy for inhibiting the activity of IspF. Allosteric inhibition may lead to the development of potent inhibitors with increased specificity [9,11,12,14].

Our group also previously synthesized enantiomerically pure DXP, MEP, [9] CDPME [10], CDPME2P [11] and MECPP [12] for their respective protein characterizations, assay development, and to determine inhibitors. Based on this in-depth study, we initiated a study to determine an IspE inhibitor. Recently, 6H-1,3 thiazine [15] compounds have been synthesized using nitro acetophenone, chlorobenzaldehyde, and thiourea and reported as novel IspE inhibitors [16]. Thiazine showed the best activity against *P. vulgaris* (3.1 µg/mL), *E. coli* (1.5 µg/mL), *P. aeruginosa* (3.1 µg/mL). However, the synthesized compounds used were racemic mixtures. They were not tested against IspE enzymes, and toxicity studies were not conducted in detail. Similarly, the recently reported IspE inhibitors [17,18] were not found to be active against mycobacterium. The inhibitors followed nine steps to synthesize the final product with low yield, and were not chiral pure compounds [19]. Hence, the role of MEP in the biosynthesis of isopentenyl diphosphate and its impact on the development of antimicrobials targeting the synthesis of this critical substrate in *M.tb*, *M. avium*, *M. abscessus* and other pathogens has high potential.

## 2. Materials and Methods

### 2.1. Materials and Strains

Tetrahydro-beta-carboline was purchased from ACROS (Morris Plains, NJ, USA). Morpholine-4-sulfonyl chloride was purchased from Oakwood Chemical (Estill, SC, USA). *Mycobacterium tuberculosis* H37Ra (*M.tb*) and *Mycobacteroides abscessus* 19977 (*M.ab*) were purchased from the American Type Culture Collection (Manassas, VA, USA). *Mycobacterium avium* (*M. avium)* was obtained from the Clinical Pathology/Microbiology Laboratory at Nebraska Medicine, Omaha, NE. 7H9 medium was purchased from BD (Sparks, MD, USA). NMR spectroscopies were recorded on a Varian Unity/Inova-500 NB (500 MHz, Varian Medical Systems Inc. Palo Alto, CA, USA). Chemical shifts are reported in parts per million (ppm) downfield from TMS. Molecular weight is determined using MALD/TOF (Applied Biosystems, CA, USA).

### 2.2. General Synthesis

Sulfonyl chloride (1.5 eq.) at room temperature was added to the mixture of tetrahydro-beta-carboline (1 eq.) and K_2_CO_3_ (3 eq.) in DMF. KI (0.3 eq.) was added to the reaction mixture after 1 h. The mixture was stirred overnight at room temperature. The mixture was washed with water, extracted with CH_2_Cl_2_, and dried over MgSO_4_. Flash column chromatography was performed to purify the crude products.

#### 2.2.1. 4-((1,3,4,9-Tetrahydro-2H-pyrido[3,4-b]indol-2-yl)sulfonyl)morpholine [1]

The reaction was stirred overnight at room temperature. The mixture was washed with water, extracted with CH_2_Cl_2,_ and dried over MgSO_4_. Flash column chromatography was performed to purify the crude products.

The residue was purified by column chromatography (silica, ethyl acetate/hexane) to give **1** as a white solid (40%). ^1^H NMR (500 MHz, CDCl_3_): δ 8.04 (s, 1H), 7.48 (d, *J* = 8.0, 1H), 7.31 (d, *J* = 8.0, 1H), 7.18 (t, *J* = 8.0, 1H), 7.12 (t, *J* = 8.0, 1H), 4.51 (s, 2H), 3.70 (t, *J* = 5.0, 4H), 3.64 (t, *J* = 5.0, 2H), 3.25 (t, *J* = 5.0, 4H), 2.87 (t, *J* = 6.0, 2H). ^13^C NMR δ:136.1, 128.9, 126.6, 122.0, 119.6, 117.9, 110.9, 108.0, 66.2, 46.2, 44.3, 43.9, 21.4. MALDI TOF *m*/*z* (M + Na)^+^; Calcd. for C_15_H_19_N_3_O_3_SNa: 344, found: 344.

#### 2.2.2. 4-((1,3,4,9-Tetrahydro-2H-pyrido[3,4-b]indol-2-yl)sulfonyl)benzo[c][1,2,5]thiadiazole [2]

The reaction was stirred overnight at room temperature. The mixture was washed with water, extracted with CH_2_Cl_2_, and dried over MgSO_4_. The residue was purified by column chromatography (silica, ethyl acetate/hexane) to give **2** as a white solid (95%). ^1^H NMR (500 MHz, DMSO-d_6_): δ 10.7 (s, NH), 8.30 (d, *J* = 8.5, 1H), 8.29 (d, *J* = 6.0, 1H), 7.85 (t, *J* = 7.5, 1H), 7.24 (d, *J* = 8.0, 1H), 7.20 (d, *J* = 8.0, 1H), 6.98 (t, *J* = 8.0, 1H), 6.86 (t, *J* = 8.0, 1H), 4.71 (s, 2H), 3.68 (t, *J* = 6.0, 2H), 2.52 (t, *J* = 5.0, 2H). ^13^C NMR δ 155.0, 149.0, 135.9, 131.9, 130.8, 130.2, 129.0, 126.6, 126.3, 121.0, 118.6, 117.5, 111.1, 106.1, 44.1, 43.6, 20.8. MALDI TOF *m*/*z* (M + Na)^+^; Calcd. for C_17_H_14_N_4_O_2_S_2_Na: 393, found: 393.

#### 2.2.3. 2-((4-Bromo-2,5-dichlorothiophen-3-yl)sulfonyl)-2,3,4,9-tetrahydro-1H-pyrido[3,4-b]indole [3]

The reaction was stirred overnight at room temperature. The mixture was washed with water, extracted with CH_2_Cl_2_, and dried over MgSO_4_. The residue was purified by column chromatography (silica, ethyl acetate/hexane) to give **3** as a white solid (58%). ^1^H NMR (500 MHz, DMSO-d_6_): δ 10.89 (s, NH), 7.36 (d, *J* = 7.5, 1H), 7.29 (d, *J* = 8.0, 1H), 7.04 (t, *J* = 7.0, 1H), 6.95 (d, *J* = 7.5, 1H), 4.60 (s, 2H), 3.71 (t, *J* = 5.5, 2H), 2.75 (t, *J* = 5.5, 2H). ^13^C NMR δ 136.0, 132.9, 132.7, 129.4, 126.4, 125.2, 121.2, 118.8, 117.8, 111.3, 109.6, 106.3, 43.9, 43.3, 21.1. MALDI TOF *m*/*z* (M + H)^+^. Calcd. for C_15_H_12_BrCl_2_N_2_O_2_S_2_: 464, found: 464.

#### 2.2.4. 2-(3-(Piperidin-1-yl)propyl)-2,3,4,9-tetrahydro-1H-pyrido[3,4-b]indole [4]

The reaction was stirred overnight at room temperature. The mixture was washed with water, extracted with CH_2_Cl_2_, and dried over MgSO_4_. The residue was purified by column chromatography (silica, 5–10% MeOH in CH_2_Cl_2_) to give **4** as a yellow oil (67%). ^1^H NMR (500 MHz, CDCl_3_); δ 9.37 (s, NH), 7.36 (t, *J* = 7.5, 2H), 7.08 (t, *J* = 7,1H), 7.02 (t, *J* = 7,1H), 3.82 (s, 2H), 2.95 (m, 8H), 2.74 (m, 4H), 2.04 (p, *J* = 7.0, 2H), 1.42 (br, 4H), 1.24 (br, 2H). ^13^C NMR δ:136.1, 130.6, 126.6, 121.3, 119.1, 117.7, 111.3, 106.8, 56.3, 54.8, 53.3, 51.4, 50.5, 49.7, 29.6, 22.9, 21.8, 20.9, 20.7. MALDI TOF *m*/*z* (M + H)^+^: Calcd. for C_19_H_28_N_3_: 298, found: 298.

### 2.3. CDPME Synthesis

Tertiary alcohol (0.54 mmol) was added to 10 mL of freshly distilled dry tetrahydrofuran (THF), and the solution was cooled to 0 °C. Potassium hydride (1.08 mmol) was added to the cooled solution, and the mixture was stirred for 30 min at room temperature by slow warming. The reaction mixture was cooled to 0 °C again, and benzyl bromide (1.35 mmol) was added slowly and warmed to room temperature. Finally, the reaction mixture was stirred for an additional 2 hr to complete the protection reaction. The reaction was quenched by the addition of a saturated ammonium chloride solution and extracted with ethyl acetate to obtain a protected product. The crude mixture was purified by silica gel column chromatography using ethyl acetate–hexane (7:3, *v*/*v*) as the solvent, yielding 0.486 mmol benzylated product MEP (90% yield). For MEP and CMP coupling, initially, CMP (0.15 mmol) was dissolved in 1 mL acetonitrile, followed by the addition of 0.6 mmol N,N, dimethyl aniline and 0.15 mmol triethylamine at 0 °C. Then, 0.76 mmol trifluoroacetic anhydride in freshly distilled acetonitrile was slowly added to the mixture. The reaction mixture was stirred for 15 min. Excess TFA and anhydride were removed under reduced pressure at low temperature. A combination of 0.45 mmol 1-methyl imidazole and 0.76 mmol triethylamine in freshly distilled acetonitrile was slowly added to the above reaction mixture, and the mixture was stirred for 30 more min. The activated CMP was added to the mixture of 0.125 mmol MEP along with activated 4 A molecular sieves in freshly distilled acetonitrile at 0 °C, and the mixture was stirred for 2 h. After the reaction was completed, the mixture was extracted with chloroform, and the aqueous layer was lyophilized to give CDP-ME. CDP-ME was purified by sequential chromatography on a Bio-Gel_ P-2 gel fine column using 100 mM aqueous ammonium bicarbonate solution followed by further purification on a benzyl DEAE cellulose anion exchange column, using a gradient of 0 to 0.5 M aqueous ammonium bicarbonate solution, resulting in a 45% yield of CDP-ME [10].

### 2.4. Determination of IC_50_ Assay

To determine the IC_50_ of compound **A1**, ADP released from the IspE catalyzed reaction was coupled to the commercially available ADP Quest HS Kinase Assay Kit (GE healthcare BioSciences Corp., Piscataway, NJ, USA) containing pyruvate kinase, pyruvate oxidase, and horseradish peroxidase to produce a fluorescent dye (resorufin). The reaction mixtures containing 50 mM Tris-Cl (pH 7.0), 100 mM ATP, 100 mM CDP-ME, 5 mM phosphatase inhibitor (Roche Bioscience, Palo Alto, CA, USA), 5 mM MgCl_2_, and 97.2 pmol Rv1011 I in a 50 mL final reaction volume in 96 well black microplates with clear bottom (Costar, New York, NY, USA) were incubated at 37 °C for 30 min. Subsequentially, 25 mL reagent A and 50 mL reagent B (ADP Quest HS Kinase Assay Kit) were added sequentially. After shaking and incubating for another 15 min at room temperature, fluorescence was measured using a Synergy^TM^ Multi-Detection Microplate Reader (BioTek instruments, Winooski, VT, USA) at an excitation wavelength of 530 nm and emission wavelength of 590 nm [10].

#### 2.4.1. Minimum Inhibitory Concentration (MIC) against *M. avium*

Ninety-six-well plates were seeded with 5 × 10^5^ CFU/mL in a volume of 200 μL of 7H9-OADC medium per well for *M.avium*. Compounds were dissolved in DMSO and serially diluted as 1:2 dilutions to give the final concentration ranging from 0.25 to 128 μg/mL per well. The final concentration of DMSO in the assay plates did not exceed 2%. For *M. avium*, the plates were incubated at 37 °C for 4–5 days, and MIC was determined by reading OD_600_. MICs were determined using a Tecan microplate reader (OD_600_) [20,21,22,23].

#### 2.4.2. Minimum Inhibitory Concentration (MIC) against *M. abscessus*

Ninety-six-well plates were seeded with 5 × 10^5^ CFU/mL in a volume of 100 μL of 7H9-OADC medium per well for *M. ab*. Compounds were dissolved in DMSO and serially diluted as 1:2 dilutions to give the final concentration ranging from 0.25 to 128 μg/mL per well. The final concentration of DMSO in the assay plates did not exceed 2%. The plate was incubated at 37 °C for 3 days. MICs were determined using a Tecan microplate reader (OD_600_) [20,21,22,23].

#### 2.4.3. Minimum Inhibitory Concentration (MIC) against *M. tuberculosis* H37Ra

A microdilution method was used to determine MIC. Ninety-six-well plates were seeded with 1 × 10^6^ CFU/mL in a volume of 200 μL of 7H9-OADC medium per well for *M.tb*. Compounds were dissolved in DMSO and serially diluted as 1:2 dilutions to give the final concentration ranging from 0.25 to 128 μg/mL per well. The final concentration of DMSO in the assay plates did not exceed 2%. The plate was incubated at 37 °C for 2 weeks, and MICs were determined using a Tecan microplate reader (OD_600_) and confirmed by resazurin reduction assay [20,21,22,23].

## 3. Results

Strategic targeting of the MEP pathway represents a promising avenue for the development of next-generation antimicrobials. As researchers unravel the complexities of this pathway, they gain insights that inform the design of inhibitors with enhanced efficacy and specificity. The potential impact of such inhibitors extends beyond conventional drug development paradigms, providing a novel approach to combating infections caused by a diverse array of human pathogens.

The absence of the MEP pathway in human cells not only underscores its appeal as a drug target but also accentuates the selectivity achievable in drug design. This selectivity is a key determinant in minimizing potential side effects on human cells while effectively targeting the microbial pathogens. The unique biochemical features of the MEP pathway, such as its reliance on specific enzymes and the absence of counterparts in mammalian cells, offer a compelling case for its exploitation in drug development.

Reported IspE inhibitors showed good IC_50_ but less MIC activity [17,18,24]. Therefore, determining a new IspE inhibitor and its lead optimization is important with high activity. The lack of MIC in the reported compounds may be due to the presence of the MEPP pathway (resistance by novel different pathway approach). We discovered a new lead compound, and lead optimization will be carried out with our novel lead IspE inhibitor. These compounds will be designed to overcome potential resistance to the MEP pathway and inform studies of new mechanistic changes in the organisms in the future [25].

A virtual high-throughput screening was carried out by docking IspE protein against 15 million commercially available compounds in the ZINC database. The top ten candidates with high docking scores were bought from commercial sources and tested against *M.tb* IspE protein in the lab. The most active compound, **A1** (Figure 2), showed enzymatic activity, IC_50_ of 6 µg/mL against *M.tb* IspE using a previously published assay and MIC of 12 µg/mL against *M.tb* (H37Rv) (Table 1), and was chosen as the lead compound for preliminary optimization. Compound **A1** has a parent heterotricyclic backbone structure. The compound also underwent a docking study with *M.tb* IspE protein and showed high binding with the carbonyl groups and IspE. Similarly high affinity was observed with the amine groups and IspE (Figure 3).

Since the active compound from the virtual screening constituted a tricyclic compound and amine groups, we set out to synthesize similar tricyclic compounds in the form of carboline derivatives. Four different carboline derivatives were synthesized (Figure 1). Briefly, carboline was alkylated or sulfated using potassium carbonate and potassium iodide at room temperature for 48 h with good yield. The synthesized, column-purified compounds were characterized by Nuclear Magnetic Resonance (NMR) and mass spectra (MS).

The synthesized compounds were screened against various bacteria. Compounds **2** and **3** showed good effect (5–10 µg/mL) against *M. avium* (Table 2). However, several compounds showed little to no effect against *M.tb,* and medium effect against *Mycobacteroides abscessus* (*M.ab*) (Table 2). These results confirm that optimization of heterotricyclic compounds could lead to an effective IspE inhibitor, and, hence, more new compounds related to virtual hit compound **A1** or compound **2** have to be synthesized and tested against mycobacterium in the future.

## 4. Discussion

The MEP pathway is an essential pathway in mycobacterium and other bacteria. It is crucial for bacterial viability. Isopentenyl diphosphate (IPP) and dimethylallyl diphosphate (DMAPP) serve as precursors for all isoprenoid compounds. There are two pathways for their biosynthesis. The mevalonate (MVA) pathway is present in eukaryotes, algae, archaea, and some Gram-positive bacteria. Gram-negative bacteria, plants, and some other Gram-positive bacteria utilize the methyl erythritol phosphate (MEP) pathway. The distinct distribution and orthogonal nature of these pathways make the MEP pathway an appealing target for antibiotics and herbicides.

Enzymes involved in MEP pathways have been characterized, and extensive research has previously been performed to develop inhibitors that block the MEP pathway [9,11,12,14]. Based on this research, we focused on identifying an IspE inhibitor that possesses 1,3-thiazine moiety, which was designed based on a lead molecule **A1** from in silico screening. Virtual high-throughput screening is a powerful computational approach employed in drug discovery to identify potential lead structures. In this method, we screened 15 million chemical compounds using a molecular docking tool which accelerates the drug discovery process by simulating the interactions between small molecules and a biological target (IspE protein) in the ZINC database. The virtual screening process generated a ranked list of compounds exhibiting the most favorable interactions with the target. These top-ranked compounds served as potential lead structures for further experimental validation. We determined MIC and IC_50_ for hit compounds and identified an active heterotricyclic compound, **A1** (Figure 4). The identified compound **A1** showed high activity against *M.tb* and enzymatic activity against *M.tb* IspE. Thus, virtual screening allowed us to design, synthesize, and test similar new heterotricyclic compounds. We synthesized carboline derivatives that possess similar tricyclic moiety. Four different carboline derivatives were synthesized (Figure 1). Briefly, carboline was alkylated or sulfated using potassium carbonate and potassium iodide with good yield. The synthesized compounds are highly active against *M. avium*. Notably, compounds **2** and **3** are active IspE inhibitors against *M. avium*, and have the potential to undergo lead optimization and in vivo study to determine a promising drug-like compound.

## 5. Conclusions

In summary, the quest to combat human pathogens has entered an era where precision and strategic innovation play pivotal roles. The MEP pathway, with its absence in mammalian cells and its central role in isoprenoid biosynthesis, stands out as a promising target for the development of antimicrobials. A virtual high-throughput screening against 15 million commercially available compounds was performed by docking IspE protein, and identified an active heterotricyclic compound, **A1**, which showed enzymatic activity; namely, IC_50_ of 6 µg/mL against *M.tb* IspE and a MIC of 12 µg/mL against *M.tb* (H37Rv). These results confirmed the importance of heterotricyclic backbone in the IspE-targeted compound. The designed, and synthesized, carboline heterotricyclic analogs observed good activity against mycobacterium and compound **2**, specifically, showed a MIC of 5 µg/mL against *M. avium*. With research endeavoring to continue unfolding the intricacies of this pathway, novel inhibitors hold the potential to revolutionize our approach to infectious diseases, offering hope for enhanced therapeutic interventions against a wide spectrum of microbial pathogens. This study provides the critical insight necessary for the potential to develop IspE inhibitors that target the MEP pathway in mycobacteria. Strategic targeting of the MEP pathway represents a promising avenue for the development of next-generation antimicrobials, with enhanced efficacy and specificity. Moreover, the virtual screening process enables the exploration of a broader chemical space, considering a vast array of compounds that may not be feasible for experimental screening. This diversity increases the likelihood of discovering novel lead structures with therapeutic potential.

## Data Availability

Data are contained within the article.

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
