# Peer review of "Investigating Novel IspE Inhibitors of the MEP Pathway in Mycobacterium"

_microorganisms, 2023, doi:10.3390/microorganisms12010018_

Round 1

Reviewer 1 Report

Comments and Suggestions for Authors

This work brings an important search about MEP inhibitors and all their importance for antimicrobial therapy.
In fact, the article is well written and presents an in-depth discussion of the molecular targets and important metabolic pathways. In my opinion, the abstract should contain more information about the results and less about the introduction of the article. As far as the introduction is concerned, it should be clearer and problematize the context more, explaining the lack of studies in the area.
Otherwise, the article is of a high standard and can be published in this journal.

Comments on the Quality of English Language

nothing to declare

Reviewer 2 Report

Comments and Suggestions for Authors

Dear Authors,

sincerly I have not yet considered the contents of your manuscript, which could be interesting, because in this form it evidences a low accuracy and an inacceptable format. Figures are entrapped in the text, which does not respect the template of MDPI. Some of them are not clear or not correctly visible.

On these considerations, I am forcet to suggest major revisions to make the paper readable and siutable for a further revision work by me.   

Reviewer 3 Report

Comments and Suggestions for Authors

Round 2

Reviewer 2 Report

Comments and Suggestions for Authors

Dear Authors,

the work has not been modifed to make Figures clearer and not intrapped in the text and therefore goes on presenting the same criticisms signalled after the first round revision. 

Author Response

The manuscript was formatted by the Journal Editorial Staff and please see the attached doc file for your convenience (I guess editorial team sent you pdf file) and hope anyone file will help. Thank you.

Reviewer 3 Report

Comments and Suggestions for Authors

The revised version adequately addresses my comments.

Author Response

Thank you.